# Motivation as a mechanism underpinning exercise-based falls prevention programmes for older adults with cognitive impairment: a realist review

Vicky Booth,[1] Rowan Harwood,[2] Jennie E Hancox,[3] Victoria Hood-Moore,[1] Tahir Masud,[4] Phillipa Logan[1]

[1]Division of Rehabilitation Ageing and Wellbeing, School of Medicine, University of Nottingham, Nottingham, UK
[2]School of Health Sciences, University of Nottingham, Nottingham, UK
[3]School of Medicine, University of Nottingham, Nottingham, UK
[4]Health Care of the Older Person, Nottingham University Hospitals NHS Trust, Nottingham, UK

**Correspondence to**
Dr Vicky Booth;
Vicky.Booth@nottingham.ac.uk

## ABSTRACT

**Objectives** This review aimed to identify mechanisms underlying participation in falls prevention interventions, in older adults with cognitive impairment. In particular we studied the role of motivation.

**Design** A realist review of the literature.

**Data sources** EMBASE, MEDLINE, CINAHL, the Cochrane Library, PsycINFO and PEDRO.

**Eligibility criteria** Publications reporting exercise-based interventions for people with cognitive impairment, including dementia, living in the community.

**Data extraction and synthesis** A 'rough programme theory' (a preliminary model of how an intervention works) was developed, tested against findings from the published literature and refined. Data were collected according to elements of the programme theory and not isolated to outcomes. Motivation emerged as a key element, and was prioritised for further study.

**Results** An individual will access mechanisms to support participation when they think that exercise will be beneficial to them. Supportive mechanisms include having a 'gate-keeper', such as a carer or therapist, who shares responsibility for the perception of exercise as beneficial. Lack of access to support decreases adherence and participation in exercise. Motivational mechanisms were particularly relevant for older adults with mild-to-moderate dementia, where the exercise intervention was multicomponent, in a preferred setting, at the correct intensity and level of progression, correctly supported and considered, and flexibly delivered.

**Conclusion** Motivation is a key element enabling participation in exercise-based interventions for people with cognitive impairment. Many of the mechanisms identified in this review have parallels in motivational theory. Clinically relevant recommendations were derived and will be used to further develop and test a motivationally considered exercise-based falls intervention for people with mild dementia.

**PROSPERO registration number** CRD42015030169.

### Strengths and limitations of this study

► Traditional systematic reviews report insufficient evidence to support falls prevention interventions among people with cognitive impairment.

► Realist review methodology explores what works for whom, in what circumstances and why (in contrast to systematic reviews which identify whether an intervention works or not), enabling exploration of contextual factors and underpinning mechanisms associated with exercise and thus the necessary conditions for participation.

► We developed a programme theory explaining the role of motivation in exercise participation, and recommendations for clinicians to support exercise components of falls intervention programmes for older adults with mild-to-moderate cognitive impairment.

► Some elements of the programme theory were not well supported by evidence, limiting the depth and detail of the recommendations, in particular around the role of exercise specifically in falls prevention.

► The cut-off date of the iterative searches and restricting to English language publications are limitations of this review, as further work may have been published that could have influenced the programme theory.

## INTRODUCTION

Falls prevention represents a complex intervention due to multifactorial causes. There is robust evidence that some interventions can reduce falls risk in the general older adult population,[1 2] but for people with dementia the effectiveness of falls prevention is uncertain.[3 4] People with cognitive impairment have a high risk of falling,[5–7] which frequently results in injury or hospital admission, but clinical guidelines cannot recommend evidence-based interventions.[8]

Exercise, at the correct dose and intensity, reduces falls risk.[1] Motivation is defined as '*the energisation and direction of behaviour*'.[9] A key challenge is how to motivate older adults to achieve sufficient exercise participation and adherence to obtain such benefits, and this is especially so for people living with dementia. Exercise interventions with older adults with dementia have reported varying levels of adherence.[10–13] People with dementia undertake less physical activity compared with those without dementia.[14] A range of factors (eg, problems with memory, executive function, carer burden and comorbidities) can influence exercise motivation.[15] However, people living with dementia populations vary according to level of impairment (mild to severe), diagnosis (eg, Alzheimer's disease, frontotemporal dementia) and support (eg, carer availability). Therefore, what is relevant to one individual might be different for another. Research is needed to explore the contextual factors and mechanisms associated with exercise engagement in older adults with dementia and to unravel some of the complexity as to what motivates whom, in what circumstances and why.

Realist synthesis is increasingly used for evaluating evidence for complex health and social interventions.[16] A realist review explores how underlying mechanisms (M) might be 'triggered' in the context (C) of a particular therapy in a particular population to produce an intended or unintended outcome (O). For example, someone with dementia who has fallen over before (C) may complete (O) an exercise programme because they are fearful (M) of falling over again. Mechanisms are further subdivided between resources, and responses.[17] Theory is generated and described through this Context-Mechanism-Outcome (CMO) heuristic.[18] CMO Configurations (CMOCs) can be linked, creating chains of possibilities and generating theories to explain why a particular outcome occurs with a specific intervention.[19] Interlinking CMOCs can be clustered together to form 'middle-range theories' (MRT), and in turn, a 'programme theory' or model of how an intervention works.[16] Realist methods encourage the incorporation of data from a range of sources, accommodating complexity that is inherent in health research.[20]

Traditional systematic reviews examine the effectiveness of a defined intervention ('*does it work?*'), as opposed to exploring the underlying mechanisms, which, in theory, may be more generalisable when studying complex interventions in heterogeneous populations. A detailed rationale for completing a realist review in this field has been published.[21 22] There are limited studies in this field, which have used different research methods.[23] Developing a theoretical framework to rationalise and explain the key principles behind an intervention will aid its development and implementation.[24]

The objective of this review was (i) to identify the underlying programme theory for participation in exercise-based falls prevention interventions in older adults with cognitive impairment, and (ii) to explore how and why that intervention reduces falls. The aim was to produce a list of recommendations that could be used clinically or to inform further intervention development.

## METHODS

### Study design

The review followed the stages identified by Pawson *et al*[25] including: i) articulating key rough programme theories to be explored, ii) searching for relevant evidence, iii) appraising the quality of evidence, iv) extracting the data and v) synthesising evidence. A detailed protocol has been published[21] and Realist And Meta-narrative Evidence Syntheses: Evolving Standards (RAMESES) guidance on publication of realist synthesis[26] reported (online supplementary table 1).

### Scoping

Initial scoping was based on prior knowledge of the falls prevention literature[23] and extensive clinical experience. The purpose was to clarify the aims of the review, develop an initial rough programme theory and direct the search strategy for the main review.

### Search processes

A phased literature search was conducted from December 2015 to June 2016. An electronic search was completed of databases: EMBASE, MEDLINE, CINAHL, the Cochrane Library, PsycINFO and PEDRO. Keywords and MESH headings were adapted according to the database used and included: *accidental falls, falls rehabilitation, falls prevention, exercise, dementia, cognitive impairment* (online supplementary table 2).

The '*iterative and interactive*'[27] search process evolved during the review, using forward and backward citation checking and manual reference list search to find additional primary evidence that was required to refine a particular aspect of the programme theory. The forward-backward citation checking was completed using Google Scholar.[28] The reference list of a systematic review[18] was manually searched. During this second search phase material was specifically chosen for i) focus on a population with dementia, ii) qualitative methods and iii) reporting experiences of completing an exercise or physical-activity intervention. The search results were screened by the researcher (VB) who documented the number of articles retrieved during each search stage using EndNote reference management software.

### Selection and appraisal of documents

Material was included regardless of study method (as is usual in realist reviews), but had to focus on an exercise intervention, be published in English and involve community-based participants or interventions. Reasons for exclusion were documented and a second researcher (PL or RH) consulted to aid decision making when required.

Titles and abstracts were screened by one researcher (VB) according to relevance of the material to the synthesis aims[29] (online supplementary table 3). Full texts

of the articles were appraised for quality in accordance with standard practice of realist reviews.[27] The relevance ('*does the research address the theory under test?'* p. 7[16]) and rigour ('*does the research support the conclusions drawn from it by the researchers?'* p. 7[16]) were assessed using a series of judgements to appraise the quality of the included studies (online supplementary table 3).

The full text for eligible studies was simultaneously assessed for quality and extraction of data by one researcher (VB). A random sample of 10% of the materials was selected and assessed by a stakeholder group comprising rehabilitation and medical clinicians and academics.

### Data extraction
Data were extracted based on relevance to the aims of the review and the rough programme theory. Data were sought that substantiated, refined or refuted the theories and described contextual characteristics. Relevant material was highlighted, labelled and recorded.[30] NVivo software and Excel was used to record and code the extracted data.

### Analysis and synthesis process
Extracted material was coded as context, mechanism or outcome and judgements regarding how this influenced the CMOCs recorded through annotations. Codes were initially allocated to each MRT within the rough programme theory, and as each article was processed, these codes were iteratively adapted according to the new material. Material that was relevant to more than one MRT were coded accordingly with links across theories.

Three waves of searching, analysis and synthesis occurred to direct the next stage of the review. Emerging findings were documented and then discussed with the stakeholder group.

### Patient and public involvement
Patients were not involved.

### RESULTS
#### Document flow diagram
The initial search identified 1954 papers (figure 1). The full text of 61 papers were eligible for screening. Sixteen papers from the initial search were not included as theoretical saturation had been reached (eg, no new findings were emerging with the consideration of new papers). The iterative search identified a further four papers.

#### Document characteristics
Twenty-one papers contributed data to the motivational mechanisms.[4 15 28 31–48] The papers varied in methodological design including: qualitative studies (n=4),[28 33 34 44] literature reviews (n=8),[4 15 32 35 36 39 43 48] randomised (n=2)[40 45] and non-randomised trials (n=4),[31 38 41 46] protocols (n=1)[42] and conference abstracts (n=2).[37 47]

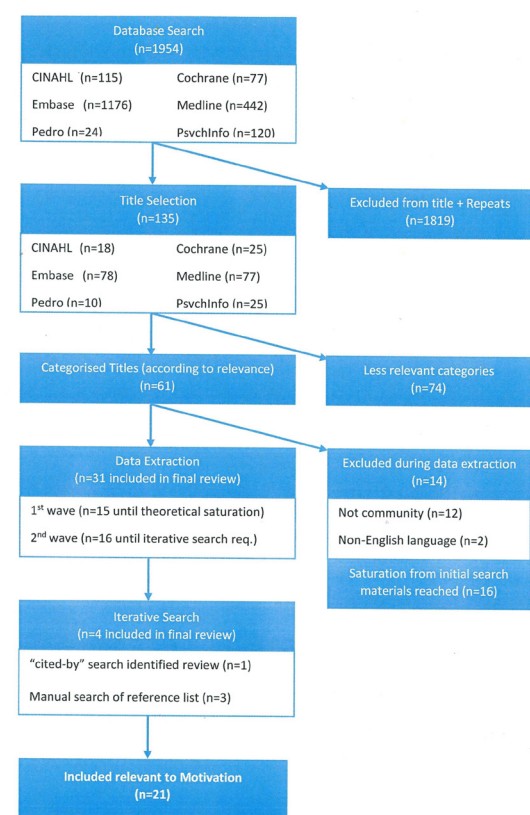

**Figure 1** Preferred Reporting Items for Systematic Reviews and Meta-Analyses flow diagram for review documents.

Contextual information, including the levels of cognitive impairment and the type, dose, and setting of the interventions, were summarised (table 1).[22]

#### Quality appraisal
All of the papers were quality appraised and critiqued according to cohesion, value, position against other material and the rough programme theory (online supplementary table 4).

#### Motivation programme theory
The programme theory elaborated from the literature comprise 11 MRTs which explain how exercise is successfully undertaken. These included 'motivation', 'physiological response', 'enjoyment', 'empowered to achieve goals', 'fearful of negative consequences', 'influenced by social or cultural beliefs', 'depression', 'tailoring of intervention', 'energy', 'quality of life' and 'function in activities of daily living'. Motivation emerged as a mechanism from many of the included studies[4 15 28 31–48] and is described in detail. Two key mechanisms were shown to be operating: a) perceived benefit and b) support.

##### 'Perception of benefit' response mechanism
The perception or feeling of benefit (box 1)[22] emerged from seven studies.[15 28 32–34 42 44] Perceiving the benefit of an exercise-based intervention could be either a response mechanism or context, depending on the individual and other context components.[28] The perception

**Table 1** Characteristics of included studies with emphasis on the intervention

| Study | Cognitive level | Type of intervention | Dose (*total hours*) | Setting |
|---|---|---|---|---|
| Pitkälä et al[45] | AD (67% moderate/ severe) | Intense, long-lasting physical exercise by dementia specialist physiotherapists in either home (HE) or group (GE) vs control (CG). | HE=1 hour, 2 p/w, 12 mth (*104*); GE=4 hours, 2 p/w, 12 mth (*416*). | Community, Finland |
| Shimada et al[47] | Amnesic MCI | Multicomponent group exercise by physiotherapists. | 90 min, 2 p/w, (n=40 sessions) 10 mth (*130*). | Community, Japan |
| Hernandez et al[41] | AD (mild/moderate) | Supervised group programme of regular physical activity | 1 hour, 3 p/w, 6 mth (*78*). | Community, Brazil |
| Hauer et al[40] | Dementia (mild/ moderate) (MMSE 21.7) | Progressive resistance and functional group training programme supervised by a qualified instructor. | 2 hours, 2 p/w, 3 mth (*52*). | Community, Germany |
| de Andrade et al[31] | AD (mild/moderate) | Aerobic, strength, flexibility, balance and cognitive exercises. | 1 hour, 3 p/w, 16 wk (*48*). | Community, Brazil |
| Garuffi et al[38] | AD | Resistance training. | 1 hour, 3 p/w, 16 wk (*48*). | Community, Brazil |
| Hüger et al[42] | Dementia (mild/ moderate, MMSE 17–26). | Progressive resistance and functional training including attention-depending motor-cognitive demands in small groups. | 2 hours, 2 p/w, 12 wk (*48*). | Community, Germany |
| Frederiksen et al[37] | AD (mild/moderate). | Aerobic exercises (exercise machines) by physiotherapist. | 1 hour, 3 p/w, 14 wk (*42*). | Community, Denmark |
| Ries et al[46] | AD (mean MMSE 23.2) | Balance exercise group with 2:1 supervision. | 45 min, 2 p/w, 8 wk (*12*). | Community, USA |
| Suttanon et al[28] | AD (mild/moderate) | Tailored home-based balance exercise by physiotherapist. | 6 visits, 6 mth (*6*). | Community, Australia |
| Cedervall et al[34] | AD (mild) | Physical activity. | Range = 'never' to 1 hour per day. | Community, Sweden |
| Cedervall and Aberg[33] | AD (mild) | Outdoor walking. | 'Routine'. | Community, Sweden. |
| Malthouse and Fox[44] | AD (MMSE 18–21) | Physical activity. | 'Routine'. | Community, UK |
| Hauer et al[39] | Dementia (moderate/ severe) | Physical training. | Range=30–150 min, 2–7 p/w, 2–30 wk (*2–525*). | Mixed. |
| Burton et al[4] | Dementia (MMSE 18.9) | Exercise or physical activity programme. | Range=1–5 p/w, 3–12 mth (*13–260*). | Mixed |
| Blankevoort et al[32] | Dementia | Physical activity. | Various. | Mixed |
| Chan et al[35] | Cognitive impairment | Physical exercise. | Various. | Mixed |
| Stubbs et al[48] | Dementia | Physical activity. | Various. | Community, mixed |
| Liu-Ambrose and Donaldson[43] | n/a | Aerobic and resistance training. | 2 p/w | n/a |
| van Alphen et al[15] | AD | Physical activity. | n/a | Community, mixed |
| Christofoletti[36] | Dementia (mixed) | Motor intervention. | Various. | Mixed |

AD, Alzheimer's disease; MCI, mild cognitive impairment; MMSE, Mini-Mental State Examination; mth, month; p/w, per week; wk, week.

of benefit could be a response mechanism that is operating when the person with cognitive impairment has '*prior experience of being active, participating in exercises and perceiving benefits of general exercise*' (p. 1180[28]) and is applicable to both the participant and carer.[28 33] Understanding an individual's previous experience of exercise and their perceptions of it can allow tailoring of approaches.[28] Perception or belief in the benefits of exercise may also

be a characteristic or feature (context) of the person with cognitive impairment or their carer, which encourages them to participate in the exercise intervention.[17]

Recognition of improvements or changes in physiological responses (eg, in physical ability) reinforces an individual's perception of benefit.[32] Identification of benefit is important for both participation and the maintenance of an intervention.[33 34] Hüger et al[42] identified that persons

with cognitive impairment can experience multiple problems which could include lack of comprehension (eg, understanding the situation). This would influence ability to identify benefits from completing exercise, and while this statement seems negative, it could be interpreted as a context component for some individuals, rather than a general characteristic of all older adults with cognitive impairment.

**Box 1    Perception of benefit Context-Mechanism-Outcome Configurations**

► An older adult ($C^1$), with dementia ($C^2$), with the belief that exercise is advantageous ($C^3$) or a positive attitude to exercise ($C^4$), will perceive the benefit ($M^{response1}$) of completing ($O^1$) an exercise intervention ($M^{resource}$).

► An older adult ($C^1$), with dementia ($C^2$), with either a carer or themselves having previous experience of being active and participating in exercise ($C^5$), and with a perception that exercise is beneficial ($C^6$), will feel encouraged ($M^{response2}$) to complete ($O^1$) an exercise intervention ($M^{resource}$).

► An older adult ($C^1$), with dementia ($C^2$), who has the ability to comprehend ($C^7$) and identifies physical or functional changes ($C^8$) will perceive the benefit ($M^{response1}$) from completing ($O^1$) an exercise intervention ($M^{resource}$).

► An older adult ($C^1$), with dementia ($C^2$), will perceive benefit ($M^{response1}$) from completing ($O^1$) an exercise intervention ($M^{resource}$) if they have an external supporter who either: understands the benefits of exercise; can prompt regarding the benefits of exercise; enables the person with dementia to compare themselves with others ($C^9$).

► An older adult ($C^1$) with dementia ($C^2$) who has a carer who perceives the benefit the person with dementia doing exercise out-weighs the risk of doing exercise or the burden of extra care duties ($C^{10}$) will provide support ($M^{response3}$) for the participation ($O^1$) in an exercise programme ($M^{resource1}$) or routine physical activity ($M^{resource2}$).

► An older adult ($C^1$) with dementia ($C^2$) who is supervised by trained personnel ($M^{resource1}$) who give clear/repeated instructions ($C^3$), optimally progress the exercises ($C^4$), provide the amount of supervision required by that individual and their needs ($C^5$) and understands the needs of persons with dementia ($C^6$) will feel supported ($M^{response1}$) to complete ($O^1$) an exercise programme ($M^{resource2}$).

► An older adult ($C^1$) with dementia ($C^2$) who is being supported ($M^{response1}$) by a professional person ($M^{resource1}$) who can time-manage ($C^7$), is knowledgeable ($C^8$), firm but encouraging ($C^9$), is kind/friendly/supportive ($C^{10}$), who understands dementia ($C^{11}$) and can develop a rapport with the individual ($C^{12}$), will do an exercise programme ($O^1$).

► An older adult ($C^1$) with dementia/AD ($C^2$) who has a carer ($C^{13}$) who receives information ($C^{21}$) and ongoing support ($C^{22}$) from the therapist/staff ($M^{resource1}$) to enable them to support ($M^{response1}$) the participation/completion of an exercise programme of the person with dementia ($O^1$).

► An older adult ($C^1$) with dementia ($C^2$) who wants to exercise in a group ($C^{23}$), will feel supported ($M^{response1}$) to complete ($O^1$) a group exercise programme ($M^{resouce1}$) or group physical activity ($M^{resource2}$).

► An older adult ($C^1$) with dementia ($C^2$) who has a poorer ability to understand and learn new information ($C^{24}$), who has not exercised previously ($C^{25}$), has ill-health ($C^{26}$) or has regular holidays ($C^{27}$), will not access the required support needed to exercise ($M^{resource1}$) and therefore will not feel supported ($M^{response1}$) to complete an exercise programme ($O^1$).

Synergy is required between carers' understanding and support, their ability to address barriers to exercise and the participants' comprehension.[15 33] Perceiving the health benefits for others also generates support and encouragement that prompts participation (eg, an individual walking his dog[33]). The feeling of encouragement could come from being able to compare themselves with others that have the same diagnosis, but only in the context where the older adult with dementia was doing well (eg, being of good health, coping with dementia symptoms).[34]

The carer's perception and belief in the benefit of exercise must outweigh the risk, care burden or adaption required to complete the exercise.[33 44] Negative connotations associated with exercising (eg, reminder of inability to do previously enjoyable activities), or adaptions or changes to routines or daily lives that are

required to support the physical activity, are destructive to the perception of benefit for both the person providing the support and the person with dementia.[33] Concern can be both facilitator and barrier to engagement in exercise, requiring a judgement between not remaining mobile and healthy, against concern about getting lost or falling.[44]

There was a range of perceived benefits from completing exercise which were not limited to health outcomes. A contentious benefit was an attempt to re-establish previous activities or the 'person' that came before the dementia diagnosis or progression of dementia.[33] Cedervall and Aberg[33] reported this perception as coming from the person providing the support. The consideration of how exercise might influence dementia or benefit falls risk was not directly reported.[44]

### Support

Support was a mechanism of motivation (box 2)[22] reported in 14 papers.[4 15 28 32–37 42 44–46 48] Support could be provided through supervision,[35 42] practical measures,[33]

strategies such as making or maintaining routines[34] or through emotional support.[34 42] There were many references to who provided the support and how it was given.

Supervision was discussed as a component of support.[35] Supervision by trained personnel '*met the special needs of persons with cognitive impairment*' (p. 153[35]) by giving clear and repeated instructions, optimally progressing the programme and providing the amount of supervision required depending on their ability to understand and learn new information.[35] Training instructors or supervisors provide more than just formal support during an intervention[42] and were influential in the commencement, participation and maintenance of exercise.[28] Key characteristics of the professional person were identified.[28] An ability to '*understand my problem*' (p. 172[44]) also emerged as important, particularly in regard to dementia. This facilitated rapport development between supporter and person with dementia, which included a relationship built on personal information[46] and trust.[15]

Carer involvement was frequently reported and was an important component regarding the support they provided.[48] The role of the carer was described by Malthouse and Fox[44] as '*facilitators to activity*' and '*gate-keepers*'. There were many ways in which carers provided support including the avoidance of stressful or negative situations,[44] providing transport,[31 41] promoting a positive attitude,[37] organising practical arrangements,[33] employing specific strategies,[34] providing additional assistance[48] and counteracting the '*loss of initiation and motivation*'[15] that were specific to that older adult with dementia and their situation. The promotion of exercise either in the practical or emotional sense, implies an underlying assumption that the activity is beneficial for the individual or themselves as a carer. However, this is a complex interaction illustrated by contradictory data. In some studies, carers received encouragement, benefit or reduced distress from providing support for the person with cognitive impairment.[36 45] In others they did not,[15 44 45 48] indicating the complexity of the caring role and the feelings associated with it.

Carers provided varying levels of support that were tailored to the individual.[28 33] Carer involvement was integral to programme delivery in one study.[28] The carer and their support was more influential the more severe the cognitive impairment and may account for why people with severe dementia were still able to engage in interventions.[48] However, it was highlighted how complex the support component is, particularly as impairments progress.[48] van Alphen *et al*[15] suggest that because persons with dementia require care and support, they are more influenced by support as a variable within an intervention. Carers themselves also required support, with information identified as a resource mechanism to enable the support to take place.[15 28]

A group setting for the intervention provided support from both the trained staff and social aspects of the group.[38 41] Individuals in the group contributed to the intervention, providing understanding of the issues and

---

**Box 3   Intervention contextual Context-Mechanism-Outcome Configurations**

► An older adult ($C^1$) with mild-to-moderate dementia ($C^2$).
► Interventions can be in either a home or group setting according to the preferences/wishes of the individual and their carer/spouse ($C^3$) (a home setting might be preferable for those wanting individual support from the intervention staff, or a group setting might be preferable for those wanting carer respite, or opportunities for social contact or engagement).
► Interventions that are multicomponent combining physical (including strength/resistance, balance, endurance/mobility, aerobic) and cognitive exercises ($C^4$), at the correct intensity and level of progression ($C^5$), supported in the correct way by suitable staff and materials (interaction, communication and connection) ($C^6$) and with consideration for speed of initiation, length of intervention, encouragement of active lifestyle and enjoyment ($C^7$).
► Intervention that is provided flexibly ($C^8$), for 6–12 months ($C^9$), 2–3 times a week ($C^{10}$), for minimum 15–20 min or whatever can be done or fit in with routine ($C^{11}$).

---

experiences of someone with dementia.[15 44] Positive results from the social aspect of a group intervention were reported by some studies directly (eg, through outcome measures)[28] and indirectly (eg, through researcher opinion).[39] However, this was not consistent across all studies.[38] Differences in participation and outcomes could be explained by the personality and preferences of the individuals. Some individuals had strong opinions on attending groups of people with dementia and this in itself will have influenced their participation.[28 40 45]

Lack of support resulted in poor adherence or participation in exercise.[4] Contexts that contributed to poor participation included lack of previous exercise experience, ill-health and holidays.[4] Lack of support is also attributed to poor results in certain interventions trials[39] and as a barrier to physical activity.[15]

### Intervention contextual characteristics

Many studies included participants with mild-to-moderate cognitive impairment (box 3),[22] with only two featuring moderate or severe dementia.[39 45] Generalising across cognitive levels is not appropriate as improvements found from an intervention at one stage may not be found in another.[39 40] A '*critical period*' for improvement within the cognitive impairment continuum was suggested.[49]

Reports conflicted regarding whether a home[28] or group[45] setting was preferable. Characteristics of the different settings suited the preferences and wishes of different individuals and their situations. A solution where multiple settings (home and group) and locations (inside and outside) for the same intervention was suggested.[15 28]

Intensity and progression of exercise were important.[4 31 46] The influence of the intervention staff (interaction, communication and connection)[28 44 46] and the materials provided[28] were considered an asset of the intervention. Particular recommendations to consider the speed of initiation and length of intervention were

identified.[44] Certain types of activity were more enjoyable for different individuals, as was the inclusion of certain types of exercise into the routine of daily life.[15 34 44 45]

The 'dose' of an intervention is a combination of frequency, duration and intensity. A range of doses was reported (table 1). Overall, the optimal dose for an exercise intervention for persons with cognitive impairment has not been defined,[38] is poorly understood,[49] but is important.[36] The concept of 'routine', both in content (such as a daily walk) and duration (such as fitting into daily life[28]) was highlighted.[15 33 34 44] A flexible approach limited absences, particularly in consideration of the mood[44] or other health conditions[28] of the participant or their carer.

## DISCUSSION
### Summary of findings
The review revealed motivation to be a core element of the programme theory underlying falls prevention interventions in older adults with cognitive impairment. Within the motivation component of the programme, two key mechanisms, perceived benefit and support, were shown to influence the extent to which an older adult with cognitive impairment is motivated to undertake an exercised-base intervention. When an older person with mild-to-moderate cognitive impairment believes that exercise will be beneficial they can use supportive mechanisms and contexts to complete an exercise programme. Support as a motivational mechanism requires a 'gate-keeper', such as a therapist or carer, who shares or takes responsibility for the perception of exercise as beneficial, thereby enabling the person with dementia to access and participate in exercise programmes. A perception of benefit is both a mechanism and contextual feature within this programme theory. Lack of access to support had a detrimental effect on adherence and participation in exercise.

### Strengths and limitations
This review progresses falls prevention research by using a novel approach. The main strength of this review is the successful completion of realist rationale in a historically positivist research field which prioritises causal probabilities over generalisability; an intervention may benefit a group on average, but we can be unsure if a given individual will benefit or be harmed. The realist review methodology was well-suited to the research question. Consideration of the mechanisms underpinning exercise-based interventions allowed development and extrapolation of the theoretical rationale. Exploring and documenting context components allows individualisation.

Transparency is encouraged in realist methods. The potential influence of the researcher in interpretation is acknowledged and, while being a potential source of bias, also assisted in the theory development and interpretations. Recognition of underlying or 'hidden' mechanisms and understanding of the CMOCs was strengthened by the experience of the main researcher (a physiotherapist) and the stakeholder group from their work with older people and falls prevention.

There are a number of limitations to this review. A micro (interpersonal) level[50] was the focus for the MRT's and overall programme theory, but the review did not consider meso (institutional) or macro (government and policy) levels of social structure.[50] The review did not base the rough programme theory on any overarching motivational theories (such as self-determination theory[51 52]). Theoretical frameworks are typically consulted to structure realist reviews (eg, search strategies and data analysis).[50 53]

The quality and content of the evidence available limited the review. Quantitative methods are more prevalent in research involving falls interventions with publications following specific reporting standards that does not encourage theoretical speculation. Information regarding participants and their influencing characteristics were rarely discussed. Greater contextual and resource information may be a product of recommendations for increased detail in reporting interventions (eg, the TIDieR guidelines[54]).

None of the included studies provided any insight into potential CMO configurations linking exercise to falls. Included studies featured both exercise and physical activities interventions. Exercise is a valuable intervention to reduce falls risk, and yet these studies did not generate theory connecting falls prevention as an outcome or motivator to exercise for people with dementia. The results of the review reflect this and are therefore limited.

Only one researcher completed the screening and data extraction. Material relevant to the review may not have been identified. The literature search was conducted in 2015 and therefore further material may have been subsequently published. Further iterative searches and snowball searching was not completed in view of the restricted time and resources. Only papers published in English were included.

### Future research directions
All of the materials included within the review described participants who had either completed regular physical activity or the exercise-based intervention under study. The perspective of those not completing an exercise-based intervention must be considered for further programme theory refinement particularly considering the motivational mechanisms.

Further research could focus on the assessment and/ or measurements of these mechanisms, for example, by investigating the assessment of perceived benefit through use of measurements or scales.

The review process has clearly directed the need for a realist evaluation to test the refined programme theory. A realist evaluation could use data from an existing exercise-based intervention in people with mild dementia and cognitive impairment to assist participation and adherence.[55 56]

**Table 2** Clinically relevant recommendations from the review results

| Focus | Recommendation |
| --- | --- |
| Who | Older adults with mild-to-moderate cognitive impairment. |
| | If a person with dementia has the belief that exercise is advantageous, a positive attitude to exercise, the ability to understand the benefits of exercise or is able to identify the physical or functional changes from doing exercise, then they will perceive the benefit of doing exercise. |
| | If a person with dementia perceives the benefit, they will participate in exercise-based intervention. |
| What | Multicomponent exercise-based intervention that: |
| | ► combines physical (including strength/resistance, balance, endurance/mobility, aerobic) and cognitive exercises. |
| | ► is appropriately intensive and progressive. |
| | ► is supported by suitable staff (who can interact, communicate and connect) and materials. |
| | ► considers speed of initiation, length of intervention, encouragement of active lifestyle and enjoyment. |
| | ► is delivered in a flexible manner for at least 15–20 min (or whatever can become or fit in with routine) 2–3 times a week for 6–12 months. |
| | ► can be delivered at home (for those wanting or needing 1:1 support from the intervention staff) or in a group (for those wanting carer respite, increase in habitual physical activity or socialising aspects). |
| Circumstances | Support can provide encouragement for completing an exercise-based intervention. |
| | Sources of support can include but are not exclusively supplied by trained intervention staff, carer, spouse, family member. |
| | If support is being provided by trained intervention staff, then they should have professional competence including: |
| | ► time-management; |
| | ► knowledgeable; |
| | ► firm but encouraging; |
| | ► kind, friendly and supportive; |
| | ► understanding of the issues experienced by persons with dementia; |
| | ► rapport development. |
| | Trained intervention staff supporting an intervention should: |
| | ► provide clear and repeated instructions. |
| | ► optimally progress the exercises. |
| | ► provide the amount of supervision required by that individual and their needs. |
| | ► understand the needs of persons with dementia. |
| | If support is being provided by a carer, then the intervention should provide information and ongoing support to enable them to continue. |
| | Carers supporting an intervention should: |
| | ► perceive and understand the benefit of the person with dementia doing exercise. |
| | ► provide transport or consider practical arrangements for access to the intervention. |
| | ► have a belief in the benefit of exercise. |
| | ► implement supportive strategies and/or assistance in the manner required by the person with dementia. |
| | If the carers or supporters perception of the benefits of doing exercise outweighs the risk, concern or burden of extra care duties, then the intervention will be encouraged. |

## Comparison with existing literature

Realist reviews have previously been undertaken to explore issues involving people living with dementia[57–61] and aid explanation in other health systems and complex interventions.[62–66]

These findings relate to a wider literature and existing theories of motivation. Many of the CMOs identified in this review have parallels with self-determination theory (SDT[67 68]). SDT is a theory of motivation which focuses on the mechanisms by which the social environment created by significant others (eg, therapists or carers) influences individuals' motivation to engage in specific behaviours (eg, exercise). Previous SDT research (eg, Murray *et al*[69]) has focused on the communication style used by healthcare professionals (eg, what they say and how they say it) and the extent to which this satisfies participants' basic psychological needs (for competence, autonomy and relatedness[68]). This review has highlighted communication strategies (eg, developing a rapport with the individual, being firm but encouraging and promoting optimal progression) similar to those considered as need-supportive, and associated with participant completion of exercise programmes. Thus, future intervention research within this population may want to consider drawing from SDT and training therapists and/or carers to adopt a need-supportive communication style.

Carer perception of benefits and support was found to be an important component of the programme theory, which is also found in existing theories of motivation. The 'perception of benefit' response mechanism has parallels with the SDT mechanism of 'identified regulation'. Another key assumption of SDT is that there are qualitatively different reasons underlying behavioural engagement.[70] One of the more autonomous reasons for engaging in behaviour is identified regulation, which is when individuals engage in an activity because they identify with the benefits. Similar to the CMOs presented within the 'perception of benefit' section of the results, research based on SDT has found that identifying with the benefits is an important mechanism mediating the

relationship between the social context created by significant others (eg, therapist or carers) and individuals' engagement in exercise behaviour.[71] Previous research has suggested that SDT is a suitable framework for investigating exercise engagement of older adults,[72] however, SDT has not been applied in research exploring exercise participation in individuals with cognitive impairment or dementia. This review extends current knowledge by highlighting a potential limitation with the applicability of the SDT construct of identified regulation to all individuals with dementia as some may not have the psychological capability or capacity to comprehend ($C^7$) the benefits of taking part in the exercise programme.

Achievement Goal Theory (AGT)[73 74] is another theory of motivation which has similarities to the findings of this review, and is also conceptually related to SDT.[75] Similar to SDT, AGT suggests that an important prerequisite for motivated behaviour is a desire to feel competent.[76] Results revealed comparison with others in a group setting as a motivating factor for older adults completing an exercise intervention but only when their performance is superior. AGT proposes that individuals can be more or less task-involved or ego-involved. Individuals who operate a more task-involved goal perspective perceive themselves as successful when they try their best and improve their own performance.[77] In contrast, more ego-involved individuals compare their performance with others and feel successful only when their performance is superior.[74] Previous research suggests that although encouraging other-referenced comparisons may be a positive motivator in the short-term, in the long-term it can be associated with maladaptive outcomes such as lower levels of exercise participation.[78] Therefore, the inclusion of comparison against others in exercise interventions should be used with caution.

Making or maintaining routines was identified as a support mechanism associated with exercise programme completion. Similarly, a meta-analysis[79] found the creation of physical activity habits to offer a means to support maintenance of physical activity behaviours overtime. Thus, future research looking to support the development of routine exercise in older adults with dementia may want to consider psychological theory on non-conscious processes, such as habit formation in order to support long-term exercise completion.

## CONCLUSION AND RECOMMENDATIONS

Older adults with mild-to-moderate cognitive impairment experience falls. Interventions, such as exercise, should be considered a resource that can positively influence an outcome of preventing falls, when used in the right circumstances or contexts. This realist review highlighted that consideration of the circumstances and underlying mechanisms for exercise-based interventions are important and could lead to greater success for future research, the individuals involved and their support networks. Recommendations for what types of exercise-based interventions for people with dementia, under what circumstances would aid motivation are provided in table 2.[22]

Benefits of exercise perceived by the carer or supporter for the person with dementia include: mood, behaviour, weight, flexibility, ageing, and enjoyment of everyday life.

**Acknowledgements** The authors would like to thank Dr Geoff Wong, of the Nuffield Department of Primary Care Health Sciences, University of Oxford, for the invaluable help with developing the protocol and the expert knowledge of realist synthesis.

**Contributors** VB completed the search, data extraction, analysis and synthesis of the review as part of her PhD (PhD in Rehabilitation and Ageing, School of Medicine, University of Nottingham). RH, VH-M, TM and PL supervised the development of the review and critically reviewed all text and theory development as part of the stakeholder group. JEH critically reviewed the findings against established motivational theories and assisted in content development. All authors contributed to the content of the review, read and approved the final manuscript.

**Funding** This paper presents independent research funded by the Alzheimer's Society, UK, with the Healthcare Management Trust through a Clinical Training Fellowship (grant number 206), and the UK National Institute for Health Research under its Programme Development Grants (RP-DG-0611-10013) and Programme Grants (RP-PG-0614-20007) for Applied Research funding scheme.

**Disclaimer** The views expressed are those of the authors and not necessarily those of the NIHR or the Department of Health and Social Care.

**Competing interests** None declared.

**Patient consent for publication** Not required.

**Provenance and peer review** Not commissioned; externally peer reviewed.

**Data sharing statement** The datasets used and/or analysed during the current study are available from the corresponding author on reasonable request.

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
