## [Reviewer comments · BMJ Open]

ARTICLE DETAILS

TITLE (PROVISIONAL)	Motivation as a Mechanism Underpinning Exercise-based Falls Prevention Programmes for Older Adults with Cognitive Impairment: A Realist Review
AUTHORS	Booth, Vicky; Harwood, Rowan; Hancox, Jennie; Hood-Moore, Victoria; Masud, Tahir; Logan, Phillipa

VERSION 1 - REVIEW

REVIEWER	Daniel Schoene Institute of Medical Physics, Friedrich-Alexander University Erlangen-Nürnberg, Germany
REVIEW RETURNED	12-Nov-2018

GENERAL COMMENTS	The manuscript entitled 'Motivation as a Mechanism Underpinning Exercise-based Falls Prevention Programmes for Older Adults with Cognitive Impairment: A Realist Review' describes a realist review focusing on factors related to adherence/participation in exercise/physical activity interventions for falls prevention in people with dementia or other types of cognitive impairments. The topic is highly relevant and fits the scope of BMJ Open. The manuscript is well written. It follows the PRISMA statement and the relevant methodology for this fairly new type of review. I have added some comments below that in my opinion, when responded to properly, will make the reading of this manuscript easier. Abstract „Publications reporting exercise-based interventions for adults with dementia (any level of cognitive impairment) in the community were“ Wouldn't it be more correct to state „in people with cognitive impairment, including dementias“? Introduction More is required than a sentence on anecdotal evidence with regard to the need of this review. Include relevant literature on adherence in cognitively impaired adults (and check for eligibility), e.g.: van der Wardt V, Hancox J, Gondek D, Logan P, Nair RD, Pollock K, Harwood R. Adherence support strategies for exercise interventions in people with mild cognitive impairment and dementia: A systematic review. Prev Med Rep. 2017 18;7:38-45. Cox KL, Cyarto EV, Ellis KA, Ames D, Desmond P, Phal P, Sharman MJ, Szoeki C, Rowe CC, Masters CL, You E, Burrows S, Lai MMY, Lautenschlager NT. A Randomized Controlled Trial of Adherence to a 24-Month Home-Based Physical Activity Program
---

and the Health Benefits for Older Adults at Risk of Alzheimer's Disease: The AIBL Active-Study. J Alzheimers Dis. 2018.

Watson E, Yu F. Monitoring exercise delivery to increase participation adherence in older adults with Alzheimer's disease. J Gerontol Nurs. 2013; 39(5):11-4.

Yu F. Improving recruitment, retention, and adherence to 6-month cycling in Alzheimer's disease. Geriatr Nurs. 2013;34(3):181-6.

Tak EC, van Uffelen JG, Paw MJ, van Mechelen W, Hopman-Rock M. Adherence to exercise programs and determinants of maintenance in older adults with mild cognitive impairment. J Aging Phys Act. 2012;20(1):32-46.

Suttanon P, Hill KD, Said CM, Byrne KN, Dodd KJ. Factors influencing commencement and adherence to a home-based balance exercise program for reducing risk of falls: perceptions of people with Alzheimer's disease and their caregivers. Int Psychogeriatr. 2012;24(7):1172-82.

McCurry SM, Pike KC, Logsdon RG, Vitiello MV, Larson EB, Teri L. Predictors of short- and long-term adherence to a daily walking program in persons with Alzheimer's disease. Am J Alzheimers Dis Other Demen. 2010;25(6):505-12.

The introduction to realist reviewing is rather theoretical and might be easier to digest with practical examples related to topic for M, C & O.

Nothing is written in the introduction section about motivation in general and motivation for exercise participation in particular.

It should be described clearly why in the authors' opinions exercise/PA interventions for falls prevention are different from other exercise/PA interventions.

Methods
Search process

Please state date of search and time frames searched in databases.

„Material was chosen for; i) focus on dementia population“
Please specify whether other types of cognitive impairment were considered too and which.

Results

Including literature reviews and single studies, how did you make sure that articles did not deal with the same trial data?

How can a protocol paper provide data/study information on the topic if no results are presented?

	From the results, it remains unclear how the review's findings relate to falls prevention rather than exercise/PA participation in general. Discussion The assessment of perceived benefit (psychometric properties, limitations) in people with dementia should be discussed briefly. One result of this review is the necessity of adequate intensity and progression levels. How was this assessed with regard to outcome participation/adherence? It should be also briefly discussed that dose is relevant to outcome fall-risk as well and there are some dose guidelines partially existent from the literature (e.g. Sherrington, BJSM 2017). Differences between exercise (balance exercise) and physical activity (walking the dog) related to review outcomes? This is also relevant as evidence with regard to falls prevention is clearer for exercise. Differences between this special population and knowledge from other populations with regard to factors influencing adherence to exercise/PA interventions should be discussed as well. References For some references first and last name have been used inconsistently. Please adhere to journal requirements.
--	---

REVIEWER	Caroline Bulsara University of Notre Dame Australia Australia
REVIEW RETURNED	04-Dec-2018

GENERAL COMMENTS	ABSTRACT: The study aim is clear but there are some issues with grammar here and in the manuscript. I would not make specific mention of self determination theory here as it is only one component of the wider findings of the review. INTRODUCTION: Further detail is required in places and sentences require expansion. For example, the first sentence noted that falls prevention is complex intervention but does not briefly state why. More detail needed by adding to this sentence. The authors demonstrate a logical approach to the review and this section is well defined. METHODS: Attention to grammar is needed here. RESULTS: Figure 1 was too small to review correctly. Generally, very thorough and well written. However, there are a few careless grammatical errors. In addition, some terminology is not well defined and needs clarification. For example, what is a 'well populated mechanism' in the first paragraph? Page 11 (Line19) - in what ways are 'context features' relevant for participant and carer? Next paragraph (Line 39/40) - perhaps be more specific with the term 'lack of comprehension'. In what ways? Page 12 - what is meant by 'older adult with dementia was doing well' - define 'doing well'. Page 14 - 'carers were frequently discussed' - in what context? What about carers was discussed? Line 32-33 of 2nd paragraph - could the authors clarify the point being made here? I would suggest adding a summary sentence.
--

	Page 15 - Lines 7 -8 - provide bracketed explanation of the direct and indirect positive results. DISCUSSION: define exactly what type of motivation you mean here - perhaps even adding to the existing sentence to provide context for the reader. STRENGTHS AND LIMITATIONS: Typo to start this section (i.e. 'review progress'). Page 19 Section 'Comparison with existing literature' needs to include mention of carers and support again. Page 20 - Line 25 onwards - I suggest explaining briefly how AGT is related to SDT. CONCLUSION: Why is "outcome" in inverted commas? Table 2: section titled - 'Carers supporting an intervention' Dot point 3. I would suggest rewording 'have a positive attitude' to indicate the importance of the carer actually believing that the intervention can work. Supp table 2 - I would have liked to see explanation of 'the value' in terms of what?
--	--

REVIEWER	Victoria Goodwin University of Exeter
REVIEW RETURNED	04-Jan-2019

GENERAL COMMENTS	Thank you for inviting me to review this excellent paper. I have a few clarifications that relate to the methods but otherwise a nice read with clear results and conclusions Page 6 line 31: cited by and manual ref list search is generally referred to as forward and backward citation checking. Usually refers to included studies only rather than all potential studies (next comment also links to this) Line 37: the reference to qualitative papers doesn't seem to fit here as you don't say until later on (line 42) that you are only including qualitative studies Page 54: this seems to indicate that it isn't just qualitative studies you will include but any study design which fits with the table in the supplementary file Page 7 Line 12: can you give more info about how the quality judgements were made Line 16: how was quality assessed?
---

VERSION 1 – AUTHOR RESPONSE

Reviewer Daniel Schoene comments to author:

9. Abstract: "Publications reporting exercise-based interventions for adults with dementia (any level of cognitive impairment) in the community were". Wouldn't it be more correct to state "in people with cognitive impairment, including dementias"?

o Revision made as per suggestion (page 2, line 10-11)

10. Introduction: More is required than a sentence on anecdotal evidence with regard to the need of this review. Include relevant literature on adherence in cognitively impaired adults (and check for eligibility), e.g.,

- o Thank you for the recommendations of additional literature to this point. Revision made to include more evidence on adherence in cognitive impaired adults (page 4)
- 11. The introduction to realist reviewing is rather theoretical and might be easier to digest with practical examples related to topic for M, C & O.
 - o Revision made to provide an example of CMOs (page 4, line 23-24)
- 12. Nothing is written in the introduction section about motivation in general and motivation for exercise participation in particular.
 - o Revision made to include definition of motivation and expanding the discussion of motivation in the introduction (page 4, second paragraph)
- 13. It should be described clearly why in the authors opinion exercise/PA interventions for falls prevention are different from other exercise/PA interventions.
 - o Revision made to provide clearer justification for exercise as an intervention to reduce falls (page 4, line 7)
- 14. Methods: Search process: Please state date of search and time frames searched in databases.
 - o Revision made – please see point 5 above
- 15. “Material was chosen for; i) focus on dementia population”. Please specify whether other types of cognitive impairment were considered too and which.
 - o Other types of cognitive impairment were not considered. To maintain clarity to the document, a list of other populations that were not considered in this iterative second phase of the search have not been included in the revision.
- 16. Results: Including literature reviews and single studies, how did you make sure that articles did not deal with the same trial data?
 - o The nature of a realist review is to generate theory. This type of review approaches the material from the perspective of what is discussed, demonstrated or learnt from the data, rather than make comment on the data itself. Therefore, it was not important within this review to exclude material that had been based on the same data set and this comment has not been included in the revision.
- 17. How can a protocol paper provide data/study information on the topic if no results are presented?
 - o Please see response to point 16 above. A protocol may provide insight into the reasoning or potential mechanisms of the intervention it is proposing to study. Therefore, this type of material was not excluded and this comment has not been included in the revision.
- 18. From the results, it remains unclear how the review’s findings relate to falls prevention rather than exercise/PA participation in general.
 - o Revision made to explain why the review does not make explicit the differentiation of exercise in general compared to exercise for falls prevention (page 18, line 15-19)
- 19. Discussion: The assessment of perceived benefit (psychometric properties, limitations) in people with dementia should be discussed briefly.

o Thank you for the comment regarding including the assessment of perceived benefit. After discussion with the other authors, the decision has been made not to include this revision into the document. The quantitative measurement of perceived benefit was not discussed in the included papers. However, this is an interesting consideration and has been included as a revision made to the 'future research directions' section (page 19, line 6-7)

20. One result of this review is the necessity of adequate intensity and progression levels. How was this assessed with regard to outcome participation/adherence? It should be also briefly discussed that dose is relevant to outcome fall-risk as well and there are some dose guidelines partially existent from the literature (e.g. Sherrington, BJSM 2017).

o Thank you for the comment regarding adding a discussion point on intensity and dose. The dose of exercise in relation to falls risk has been included in the introduction, as per comment 13 (page 4, line 7)

21. Differences between exercise (balance exercise) and physical activity (walking the dog) related to review outcomes? This is also relevant as evidence with regard to falls prevention is clearer for exercise.

o Revision made to highlight both exercise and physical activities were included in the review (Page 18, line 16)

22. Differences between this special population and knowledge from other populations with regard to factors influencing adherence to exercise/PA interventions should be discussed as well.

o Thank you for the comment regarding including the assessment of perceived benefit. After discussion with the other authors, the decision has been made not to include this revision into the document as the 'comparison with existing literature' section extensively reports motivational theories which are not explicitly involving people with dementia.

23. References: For some references first and last name have been used inconsistently. Please adhere to journal requirements.

o Revisions made to the reference list for consistency with journal requirements

Reviewer Caroline Bulsara Comments to Author:

24. ABSTRACT: The study aim is clear but there are some issues with grammar here and in the manuscript. I would not make specific mention of self-determination theory here as it is only one component of the wider findings of the review.

o Revision made to remove self-determination theory from the abstract (page 3). Revisions made to grammar throughout.

25. INTRODUCTION: Further detail is required in places and sentences require expansion. For example, the first sentence noted that falls prevention is complex intervention but does not briefly state why. More detail needed by adding to this sentence. The authors demonstrate a logical approach to the review and this section is well defined.

o Revision made to expand why falls prevention is complex (page 4, line 2)

26. METHODS: Attention to grammar is needed here.

o Revisions made to grammar throughout the methods section.

27. RESULTS: Figure 1 was too small to review correctly. Generally, very thorough and well written. However, there are a few careless grammatical errors. In addition, some terminology is not well defined and needs clarification. For example, what is a 'well populated mechanism' in the first paragraph?
- o Revisions made to the results to clarify terminology. Revisions made to grammar throughout.
28. Page 11 (Line19) - in what ways are 'context features' relevant for participant and carer?
- o Revision made for clarity on this point (page 11).
29. Next paragraph (Line 39/40) - perhaps be more specific with the term 'lack of comprehension'. In what ways?
- o Revision made with provision of example (page 11, line 17).
30. Page 12 - what is meant by 'older adult with dementia was doing well' - define 'doing well'.
- o Revision made with provision of example (page 12, line 3-4).
31. Page 14 - 'carers were frequently discussed' - in what context? What about carers was discussed?
- o Revision made to clarify (page 14, line 6-7).
32. Line 32-33 of 2nd paragraph - could the authors clarify the point being made here? I would suggest adding a summary sentence.
- o Revision made to clarify.
33. Page 15 - Lines 7 -8 - provide bracketed explanation of the direct and indirect positive results.
- o Revision made with bracketed explanation (page 15, line 4-5).
34. DISCUSSION: define exactly what type of motivation you mean here - perhaps even adding to the existing sentence to provide context for the reader.
- o Revision made to provide greater clarity to the reader on the meaning of motivation identified within the review (page 17, line 3-7)
35. STRENGTHS AND LIMITATIONS: Typo to start this section (i.e. 'review progress').
- o Revision made (page 17, line 15)
36. Page 19 Section 'Comparison with existing literature' needs to include mention of carers and support again.
- o Revision made to highlight the carers and support in the motivational theories discussed (Page 20, line 2-4)
37. Page 20 - Line 25 onwards - I suggest explaining briefly how AGT is related to SDT.
- o Revision made to include an explanation highlighting how competence is a key concept within both SDT and AGT (page 20, lines 18-20)
38. CONCLUSION: Why is "outcome" in inverted commas?
- o Revision made to remove brackets and clarify point (page 21, line 14)

39. Table 2: section titled - 'Carers supporting an intervention' Dot point 3. I would suggest rewording 'have a positive attitude' to indicate the importance of the carer actually believing that the intervention can work.

o Revision made to reword (page 22, 3rd bullet point in “carers supporting an intervention should;” section)

40. Supp table 2 - I would have liked to see explanation of 'the value' in terms of what?

o Revision made to supplementary table to provide additional explanation of “value”

Reviewer Victoria Goodwin Comments to Author:

41. Page 6: line 31: cited by and manual ref list search is generally referred to as forward and backward citation checking. Usually refers to included studies only rather than all potential studies (next comment also links to this)

o Revision made to refer to “cited-by” as forward and backward citation checking has been made (page 6, line 13-16)

42. Line 37: the reference to qualitative papers doesn't seem to fit here as you don't say until later on (line 42) that you are only including qualitative studies

o Revision made to clarify that the qualitative method restriction was only in place for the iterative (second) search phase (page 6, line 17)

43. Page 54: this seems to indicate that it isn't just qualitative studies you will include but any study design which fits with the table in the supplementary file

o See point 42 above

44. Page 7: Line 12: can you give more info about how the quality judgements were made

o See point 6 above

45. Line 16: how was quality assessed?

o See point 6 above

VERSION 2 – REVIEW

REVIEWER	Daniel Schoene Friedrich-Alexander University Erlangen-Nürnberg, Germany
REVIEW RETURNED	22-Feb-2019

GENERAL COMMENTS	The revised manuscript entitled 'Motivation as a Mechanism Underpinning Exercise-based Falls Prevention Programmes for Older Adults with Cognitive Impairment: A Realist Review' is interesting to read and provides novel insight into exercise intervention in people with cognitive impairments. The authors have responded to the reviewer comments in sufficient detail. Few minor suggestions below. However, I do not have to review this manuscript again.
--

	Highlights „We developed a programme theory explaining the role of motivation in exercise participation, and recommendations for clinicians to support exercise components of falls intervention programmes for older adults with mild cognitive impairment.“ Why “mild”? Introduction Reference 1 only deals with exercise interventions and this should become clear from the sentence. Unclear why reference 2 from 2007 is used to state that evidence on falls prevention and dementia is uncertain when more recent meta-analytic evidence by Burton et al. 2015 exists and was used later in the manuscript. “Traditional systematic reviews examine the effectiveness of a defined intervention (‘does it work?’), as opposed to exploring the underlying mechanisms [...] Isn’t meta-regression for instance a method that investigates underlying mechanisms. An example is Sherrington et al. that you cited. Methods “i) focus on a population with dementia” From Table 1 it appears that several of the papers included were done in non-demented populations. Results “Sixty-one papers were screened for inclusion” Do you mean as full text? Please specify as all 1954 were screened at least in TIAB stage. Was theoretical saturation also confirmed / discussed by/with stakeholder group? Boxes 1-3 legend should explain abbreviations. Discussion It would be interesting to know more specifically about the perception of benefit of exercise as this may have differed between studies. Is it possible to elaborate on this topic? Do exercise participants perceive the benefit of reducing falls, increasing function, mobility, self-efficacy, finding social networks, etc.? Please add to the limitations that only English papers were considered. I also still believe that self-report of demented participants is a potential bias due to their cognitive state. This could limit the conclusions of the review. “Carer perception of benefits and support was found to be an important component of the programme theory. ” I do not see how this sentence as a stand-alone contributes to the discussion on existing published evidence.
--	--

REVIEWER	Victoria Goodwin University of Exeter, UK
REVIEW RETURNED	21-Feb-2019

GENERAL COMMENTS	Nice paper on an important topic. I am happy with the revisions
---

VERSION 2 – AUTHOR RESPONSE

Reviewer Daniel Schoene comments to author:

The revised manuscript entitled 'Motivation as a Mechanism Underpinning Exercise-based Falls Prevention Programmes for Older Adults with Cognitive Impairment: A Realist Review' is interesting to read and provides novel insight into exercise intervention in people with cognitive impairments. The authors have responded to the reviewer comments in sufficient detail. Few minor suggestions below. However, I do not have to review this manuscript again.

5. "We developed a programme theory explaining the role of motivation in exercise participation, and recommendations for clinicians to support exercise components of falls intervention programmes for older adults with mild cognitive impairment." Why "mild"?

- Thank you for highlighting the inaccuracy in this statement, the programme theory was developed from studies that included mild to moderate cognitively impaired older adults, therefore a revision has been made (page 3, line 21)

Introduction

6. Reference 1 only deals with exercise interventions and this should become clear from the sentence.

- Revision made to sentence and additional reference included (page 4, line 6)

7. Unclear why reference 2 from 2007 is used to state that evidence on falls prevention and dementia is uncertain when more recent meta-analytic evidence by Burton et al. 2015 exists and was used later in the manuscript.

- Revision made to include more recent reference (page 4, line 5 onwards)

8. "Traditional systematic reviews examine the effectiveness of a defined intervention ('does it work?'), as opposed to exploring the underlying mechanisms [...]" Isn't meta-regression for instance a method that investigates underlying mechanisms. An example is Sherrington et al. that you cited.

- Thank you for the comment. The term "mechanism" referred to within realist evaluation has a slightly different interpretation to the mechanistic interference from variables identified from a meta-regression. No revision made.

Methods

9. "i) focus on a population with dementia" From Table 1 it appears that several of the papers included were done in non-demented populations.

- Thank you for the comment. Yes some of the included papers described their sample populations as "mild cognitive impairment" or "cognitive impairment" or did not have a sample to

describe. Dementia is used in the broad term to include all types of dementia, including Alzheimer's disease and mild cognitive impairment. No revision made.

Results

10. "Sixty-one papers were screened for inclusion" Do you mean as full text? Please specify as all 1954 were screened at least in TIAB stage.

- Sixty-one papers were eligible for full text screening. Revision has to been made (Page 8, line 15-16).

11. Was theoretical saturation also confirmed / discussed by/with stakeholder group?

- Yes theoretical saturation was discussed and confirmed with the stakeholder group. No revision made.

12. Boxes 1-3 legend should explain abbreviations.

- Abbreviations have been added to the List of Abbreviations (Page 24, line 2-4)

Discussion

13. It would be interesting to know more specifically about the perception of benefit of exercise as this may have differed between studies. Is it possible to elaborate on this topic? Do exercise participants perceive the benefit of reducing falls, increasing function, mobility, self-efficacy, finding social networks, etc.?

- Thank you for the comment. Yes this would be an interesting aspect to explore in more detail, however, with this being a review the authors were unable to gain primary data on this topic. No revision made.

14. Please add to the limitations that only English papers were considered.

- Revision has been made (Page 20, line 1-2)

15. I also still believe that self-report of demented participants is a potential bias due to their cognitive state. This could limit the conclusions of the review.

- Self-report is a recognised issue within research involving people with dementia, however, this has not been included as a limitation within this review as not all the outcomes were self-reported. No revision made.

16. “Carer perception of benefits and support was found to be an important component of the programme theory.” I do not see how this sentence as a stand-alone contributes to the discussion on existing published evidence.

- A revision has been made that associates the finding from this review being linked with existing motivational theories (Page 21, line 7)